# Trajectories of Teleworking via Work Organization Conditions: Unraveling the Effect on Work Engagement and Intention to Quit with Path Analyses

**Annick Parent-Lamarche** [1,*] and **Alain Marchand** [2]

1   Department of Human Resources Management, Université du Québec à Trois-Rivières, Trois-Rivières, QC G8Z 4M3, Canada
2   School of Industrial Relations, University of Montreal, Montreal, QC H3C 3J7, Canada
*   Correspondence: annick.parent-lamarche@uqtr.ca

**Abstract:** Several countries are currently experiencing worker shortages. In this context, which favors employees, employers must improve their offer to attract and retain employees, not only in regards to wage but also in regards to work organization conditions. Teleworking is one work organization condition (or human resource management practice) that is receiving increasing attention due to its increased prevalence in recent years. This cross-sectional study's objective was to verify the influence of teleworking on work engagement and the intention to quit through its effects on work organization conditions (e.g., social support, workload, recognition, skill utilization, and number of hours worked). This study was based on the demands-resources model as teleworking can represent a demand or a resource and is likely to influence work organization conditions. Path analyses were carried out using Mplus software. A sample of 254 French Canadian staff members ($n = 254$) from 19 organizations (small and medium-sized). The results indicate that teleworking is indirectly associated with a higher level of work engagement through its effect on skill utilization. Moreover, teleworking is indirectly and negatively associated with the intention to quit through its impact on skill utilization and work engagement. More specifically, teleworking is associated with an overall lower intention to quit. This study aimed to shed light on the mechanisms underlying the associations between teleworking, work engagement, and the intention to quit. Considering work organization conditions in this sequence modifies the effect of teleworking on both outcomes. Although it can be harmful (i.e., negatively associated with work engagement) when the work organization conditions are not considered, its positive influence on skill utilization reverses this effect. From a practical perspective, it seems crucial to ensure that teleworkers can use their skills to promote the success of its implementation.

**Keywords:** teleworking; work engagement; intention to quit; skill utilization; work organization conditions; job demands–resources model; path analyses

## 1. Introduction

Along with the anticipated recession, continued inflation, layoffs, and unemployment, demographic shifts and aging populations are causing an ongoing shortage of workers in many countries [1]. In this situation, which favors employees, employers must improve their offer to them, both in terms of salary and work organization conditions. This is mandatory to succeed, considering the labor shortage. Further, it was recently reported that nearly 50% of employees in Canada plan to change jobs in 2023 [2]. To prevent the loss of employees, it is important to deepen our comprehension of the causes of employees' work engagement and intention to quit. In fact, work engagement must be a central consideration for efficient human resource (HR) management in workplaces [3,4]. Work engagement makes reference to an optimistic, rewarding, job-related mindset related to vigor, dedication, and absorption [5]. High levels of energy and mental resilience at work indicate vigor, while being fiercely involved and having a sense of significance, ardor, and

challenge reflect dedication. Absorption pertains to being intensely concentrated as well as joyfully immersed at work. Here, we concentrate on vigor and dedication, as they seem to be the central dimensions (the ideal measure) of work engagement [6] and have been linked to employees' intentions to quit [7–10]. Intention to quit refers to an employee's subjective evaluation of the probability that he or she will quit his or her job [11], and it is considered a critical precursor of actual turnover [12]. Engaged workers have an optimistic state of mind and limited time or space for pessimistic thoughts, such as thinking about quitting their job [13], which is why organizations should seek to enhance the engagement of their employees.

In this regard, a work organization condition (or a human resource management practice) that has received considerable attention is teleworking due to its increased prevalence in recent years. Teleworking (TW) refers to executing tasks elsewhere (i.e., not in the physical workplace) while still connected to it with telecommunication tools or computer-based technology [14]. While different types of teleworking exist (e.g., home-based teleworking, satellite offices, neighborhood work centers, and mobile work [15]), in this study, teleworking refers to working from home. Teleworking has long been considered an advantageous working condition that could fit into a global compensation plan or improve an employer's offer to attract and retain employees [16,17]. Moreover, the COVID-19 pandemic has perpetuated and even accelerated this organizational practice. For example, one-third of Canadian workers were involved in teleworking at the start of 2021, compared to only 4% before the pandemic [18]. It is expected that teleworking could become normal for 20–25% of workers worldwide [19], which represents a huge challenge for the working world. Consequently, research on the effects of teleworking has intensified. However, few studies have explored the health effects of teleworking [20,21]. As recently stated by Buomprisco et al. [22], health problems linked to teleworking will emerge, and research in this area will be increasingly important in the future. Moreover, there are few studies about the impacts of teleworking on job performance outcomes [9]. As a result, little is known about the underlying mechanisms of these effects. How does teleworking impact work engagement (WE) and intention to quit (IQ)? Does teleworking influence other work organization conditions, also called psychosocial risks, that are responsible for health, well-being, and performance at work?

Answering these questions is essential because teleworking is prolonged and its many benefits for employees have long been recognized (e.g., saving time commuting, better work–life balance, cost-effectiveness, lower level of stress, higher freedom, and a convenient workspace), despite certain disadvantages (e.g., interpersonal/professional reclusiveness) [15]. For organizations, teleworking is also advantageous in terms of higher employee productivity, lower absenteeism, and lower turnover, but it does result in a reduction in informal interaction and work coordination [15]. On a societal scale, teleworking could also be eco-friendly, as it reduces pollution and the carbon footprint resulting from mobility [23]. Fully 80% of new pandemic-related teleworkers indicated that they would prefer to perform a minimum of 50% of their hours from home after the pandemic [18]. Considering all this information, it seems that the trend toward teleworking will only grow stronger over time.

### 1.1. Aim of the Study

This study was cross-sectional and aimed to investigate the direct impacts of TW (i.e., teleworking from home) on work organization conditions as well as verify the indirect impacts of TW on WE and IQ through work organization conditions. To do so, 254 French Canadian staff members from 19 organizations (small and medium-sized) were studied. The identification of the underlying mechanisms that link teleworking to work engagement and intention to quit is crucial to deepening our understanding and ensuring teleworking's positive effects on employees as well as the organizations that employ them. Currently, there are several gaps in the available knowledge regarding the impacts of TW on those outcomes, significantly limiting the capacity to act effectively to prevent negative issues

from arising. The scope and originality of our study relate to the need—particularly from a practical point of view to support organizations in prevention and awareness—to develop knowledge on teleworking from the perspective of mutual benefits (employees/employers).

By focusing on the underlying mechanisms that link teleworking to work engagement and intention to quit, the current study adds to the actual literature in three ways. First, the study examines the effect of teleworking on the work organization's conditions. This is important for improving the implementation of teleworking to maximize its positive effects and reduce its negative ones. Second, it develops a comprehensive model of the mechanisms behind the association linking TW, WE, and IQ. Third, it answers research questions based on a sample of 19 small and medium-sized organizations (SMOs) rather than focusing on a single organization or industry. Indeed, the reality of employees in SMOs can be different than that of employees of other types of organizations, as they might be more habituated to interact more frequently with people and to share spatial space; the management style is also more proximal [24], and informal communications are omnipresent [25]. It is possible that the effects of teleworking will not be the same for these employees. However, there can surely be exceptions depending on the types of jobs in these organizations. Therefore, it is important to study the effects of teleworking in this specific context to propose practical implications that are specific and adapted to SMOs.

*1.2. Theoretical Model*

To achieve these contributions, we fine-tune and extend the job demands–resources (JD-R) model. This model classifies work organization conditions in terms of demands or resources. Demands at work are characterized by the physical, psychological, social, or organizational elements of the work, which are associated with certain psychological costs [26]. For their part, resources at work correspond to the physical, psychological, social, and organizational aspects that reduce the psychological costs associated with them [26]. We will attempt to verify whether teleworking is a demand and/or a resource that leads to higher or lower work engagement and the intention to quit through its effects on other work organization conditions (i.e., other demands and resources at work). The model proposes that demands and resources at work act on employees through two alternative processes: (1) Demanding jobs can lead to health problems such as exhaustion or impaired work engagement (health impairment hypothesis); (2) Resourceful jobs have motivational potential (motivational hypothesis) and are conducive to higher performance at work (e.g., low intention to quit). In addition, resources at work (e.g., work organization conditions) are known to be associated with work engagement [27]. In fact, resourceful jobs (e.g., based on decision authority, skill utilization, low workload, social support, recognition, regular work schedules, and a reasonable number of hours worked per week) motivate efforts toward task completion [28].

The theoretical model that we propose presumes that teleworking might influence other work organization conditions (also termed demands and resources at work), which is our first extension of the JD-R model. See Figure 1. In fact, teleworking could be an upstream demand or a resource that could impact other demands and resources at work. Recent studies seem to indicate that teleworking is altering the work organization's conditions. For instance, teleworking has been associated with higher job decision latitude (i.e., autonomy and skill development) [29]. Teleworking is also associated with more freedom in working hours, fewer disturbances and interruptions, and an inferior quality of relationships with colleagues [30]. According to Pulido-Martos et al. [31], the relationship between teleworking and vigor at work is lower when teleworking, even if teleworkers perceive a similar amount of support from supervisors and colleagues as their colleagues who work in the office. Thus, teleworking attenuates the positive effect of social support. Another study found that employees teleworking more days a week reported a lower level of colleagues' support, which was successively linked with increased levels of emotional exhaustion, cynicism, and cognitive stress complaints as well as lower work engagement [32]. In turn, those effects increased psychosomatic health complaints [30]. We did not find any research

that tested how job recognition could be affected by teleworking. That said, it was recently demonstrated that recognition is favorable for psychological distress, and this impact is higher in the office [33]. In other words, it appears that job recognition is passed on or noticed more easily in the official workplace compared to when teleworking. On the contrary, it was recently found that teleworking during lockdowns improved working conditions, especially work schedules, due to more comfortable workloads, reduced working hours, a compressed workweek, and flexible shift systems [34]. In sum, the effects of teleworking appear to be equivocal and merit more attention and investigation. All things considered and in conformity with the JD-R model, we suggest the following general and exploratory hypotheses:

**H1.** *Teleworking is directly associated with work organization conditions.*

Additionally, our theoretical model holds that the influence of teleworking on other work organization conditions will affect work engagement, which will successively affect employees' intentions to quit. Therefore, our second extension to the JD-R model is to propose that the link between TW, WE, and IQ might be mediated (instead of direct) by other work organization conditions, such as decision authority, skill utilization, workload, social support, recognition, an irregular work schedule, and the weekly working hours. Drawing on conservation of resources (COR) theory [35], teleworking could help to prevent a cycle of loss (i.e., a loss spiral) by influencing other work organization conditions and, ultimately, their effects on work engagement and the intention to quit. Indeed, it has been shown that work organization conditions are linked to WE [36–39] and IQ [40–43]. We also know from a previous study that teleworking is linked to lower WE and higher IQ through its impact on WE [9]. However, the impact of teleworking on work organization conditions has received little attention, along with the subsequent effects on work engagement and the intention to quit. Could the other work organization conditions in this process modify the ultimate impact of teleworking, previously identified, on work engagement and the intention to quit? Consequently, we suggested the following hypotheses:

**H2** *. Teleworking is indirectly associated with work engagement via work organization conditions.*

**H3.** *Teleworking is indirectly associated with the intention to quit through its effect on work organization conditions and work engagement.*

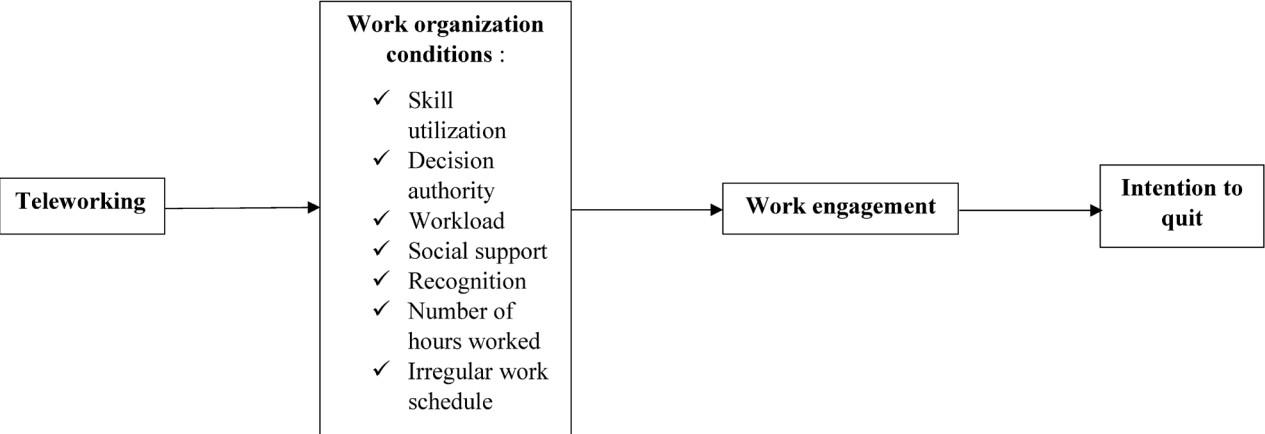

**Figure 1.** Conceptual model.

## 2. Methods

### 2.1. Procedure and Participants

The data collection took place from 1 June 2020 to 14 December 2021 within 19 Canadian organizations (the final sample included 254 employees) with the assistance of the public affairs service of the Université du Québec à Trois-Rivières during the pandemic, which made the data collection process longer and more difficult. Consequently, this study

was cross-sectional. However, the sampling method, which included 19 different SMOs, is valuable. After establishing contact, we met with HR managers and company executives to explain the implications and aims of the research. Participating organizations were given feedback in the form of a personalized profile ("HR Profile") of their employees' viewpoints on several dimensions of the work organization's conditions and human resources management practices. The aim was to help the organizations realign their practices (if needed) to better meet the needs and perceptions of their employees. The strictest ethical rules for research were followed in this study. The participants were provided with information regarding confidentiality and signed an informed consent document before completing the questionnaire (hardcopy and online versions were accessible). No monetary reward was granted, except for a $50 gift card drawn from participating staff members/employees. In this research, 19 workplaces were from the secondary (36.8%) and tertiary (63.2%) economic activity sectors (e.g., manufacturer, non-profit organization), and 21.1% were unionized. The average size was 24 employees per organization. For each of them, all staff members were qualified to complete a survey (final response rate: 74.7%). The ultimate sample was 51.2% female, with a mean age of 41.7 years and a mean household income between $60,000 and $79,999.

### 2.2. Measures

#### 2.2.1. IQ

*IQ* was assessed with a three-item, seven-point additive scale, including responses to all items (e.g., "I'm thinking about quitting my job"; $\alpha = 0.91$) covering from 1 (very strongly agree) to 7 (do not at all agree) [44].

#### 2.2.2. WE

*WE* was assessed with the Utrecht Work Engagement Scale shortened version (UWES-6) [5], which contains six items measured on a seven-point additive scale, including responses for all items (e.g., "I am enthusiastic about my job"; $\alpha = 0.91$) ranging from 0 (never) to 6 (daily). Previous studies confirmed that merging a two-dimensional (vigor and dedication) scale into one general score is a valid choice for academic analysis [6,45–48].

#### 2.2.3. TW

*TW* was assessed with one question (i.e., "Which statement best describes how you perform your work during the COVID-19 crisis?") and was coded as 0 ("I go to my usual place of work") or 1 = ("I work from home").

#### 2.2.4. Work Organization Conditions

Work organization conditions comprise diverse variables. Some of these variables (i.e., skill utilization, decision authority, and social support) were assessed using the Job Content Questionnaire [49]. Responses were measured on a four-point Likert scale ranging from 1 (strongly disagree) to 4 (strongly agree). Skill utilization was comprised of six items (e.g., "I have the opportunity to develop my own special abilities"; $\alpha = 0.72$). Decision authority consisted of three items (e.g., "On my job I have the freedom to decide how I do my work"; $\alpha = 0.77$). Social support at work was comprised of eight items (e.g., "My supervisor pays attention to what I'm saying"; $\alpha = 0.88$). To measure workload and recognition, the Effort-Reward Imbalance Questionnaire [50] was used. Each variable was measured on a four-point Likert scale ranging from 1 (strongly disagree) to 4 (strongly agree). Workload was composed of five items (e.g., "I have many interruptions and disturbances while performing my job"; $\alpha = 0.79$). Recognition included five items (e.g., "I receive the respect I deserve from my colleagues"; $\alpha = 0.84$). Number of hours worked was obtained by summing the hours worked per week. Finally, irregular work schedule was evaluated based on a single item (i.e., "In your present job, are you exposed to irregular or unpredictable work schedule?"), which was coded either as 0 ("No") or 1 = ("Yes").

### 2.2.5. Control Variables

Variables linked with work engagement and/or intention to quit were identified in past studies, including age [51–53], gender [51,53,54], educational level [55,56], household income [57], marital status [51,53], and parental status [58]. Additionally, the stress related to the COVID-19 pandemic was controlled for because data were collected during the pandemic and past research established that stress is linked to performance at work [59].

Age was computed in terms of years. Gender was computed as 0 ("Male") or 1 ("Female"). Marital status was computed as 0 ("Single") or 1 ("Living as a couple"). Parental status was computed as 0 ("No") or 1 ("Minor children [under 18 years of age] living with the respondent"). Stress related to the COVID-19 pandemic was evaluated with one question (i.e., "How has the COVID-19 crisis affected your stress level?") and was computed as 0 ("The COVID-19 crisis decreased my stress level or did not change my stress level") or 1 ("The COVID-19 crisis increased my stress level").

### 2.3. Analysis Plan

In order to empirically verify our hypotheses, we conducted path analyses with Mplus software [60] based on the approach of Preacher and Hayes [61]. With this software, we executed path analyses corrected for design effects while at the same time considering the non-independence of the data (employees nested in organizations). These analyses are theoretically useful, as they enable verifying the underlying mechanisms through which independent variables can be directly and indirectly associated with a dependent variable. In other words, they are considered a kind of structural equation modeling (SEM), which permits inferring and verifying a sequence of causal associations among different variables [62]. Path analyses are appropriate for retrieving a sharpened interpretation of the processes and hidden channels of a particular reality. More precisely, Preacher and Hayes [61] methodology allowed to verify if the association between TW, WE, and IQ was mediated by staff members' perceptions of their work organization conditions. Analyses were accomplished using a single model that calculated the direct effects of teleworking on work organization conditions (H1) as well as the indirect effects of teleworking on work engagement through work organization conditions (H2). In addition, the indirect effects of teleworking on intention to quit through work organization conditions and work engagement were analyzed (H3). To determine the significance levels of the joined variables and for each of them individually, a two-tailed probability was considered for rejection of the null hypothesis ($p \leq 0.05$). The models were tested with maximum likelihood estimation using robust standard errors (MLR estimation). The goodness of fit was assessed using the comparative fit index (CFI) and the Tucker–Lewis index (TLI). Values above 0.95 for the CFI and TLI indicate an excellent fit [63].

## 3. Results

Table 1 presents the descriptive statistics (mean/proportion, standard deviation) and correlations.

Table 2 presents the main effect and results of TW on work organization conditions (H1). The results show that TW was significantly and positively associated with skill utilization and social support and was significantly and negatively associated with weekly working hours (H1). Additionally, Table 2 presents the indirect effects of TW on both WE through its effects on work organization conditions (H2) and IQ through its effects on work organization conditions and WE (H3). TW was associated with a significantly higher level of WE through its effect on skill utilization (H2) and a lower IQ through its effect on skill utilization and WE (H3). Note that even though we did not formulate specific hypotheses regarding the effects of TW on WE and IQ, the results showed that TW was significantly associated with lower WE ($\beta = -2.312$; $p \leq 0.05$) as well as indirectly associated with a lower level of IQ via its effect on WE ($\beta = 0.724$; $p \leq 0.05$). Moreover, WE was significantly associated with a lower IW ($\beta = -0.313$; $p \leq 0.01$).

**Table 1.** Descriptive and correlational statistics.

| | M | SD | 1. | 2. | 3. | 4. | 5. | 6. | 7. | 8. | 9. | 10. | 11. | 12. | 13. | 14. | 15. | 16. | 17. |
|---|---|---|---|---|---|---|---|---|---|---|---|---|---|---|---|---|---|---|---|
| 1. | 6.62 | 4.70 | 1 | | | | | | | | | | | | | | | | |
| 2. | 25.49 | 7.20 | −0.49 ** | 1 | | | | | | | | | | | | | | | |
| 3. | 0.63 | | 0.07 | −0.08 | 1 | | | | | | | | | | | | | | |
| 4. | 18.03 | 2.82 | −0.24 ** | 0.28 ** | 0.25 ** | 1 | | | | | | | | | | | | | |
| 5. | 9.33 | 1.70 | −0.17 ** | 0.23 ** | 0.09 | 0.46 ** | 1 | | | | | | | | | | | | |
| 6. | 11.88 | 3.03 | 0.03 | −0.06 | 0.11 | 0.21 ** | 0.09 | 1 | | | | | | | | | | | |
| 7. | 26.13 | 3.76 | −0.22 ** | 0.27 ** | 0.19 ** | 0.39 ** | 0.38 ** | −0.10 | 1 | | | | | | | | | | |
| 8. | 16.69 | 2.59 | −0.26 ** | 0.28 ** | 0.13 * | 0.30 ** | 0.42 ** | −0.15 * | 0.72 ** | 1 | | | | | | | | | |
| 9. | 36.76 | 6.13 | −0.06 | 0.07 | −0.16 ** | −0.03 | −0.14 * | 0.26 ** | −0.16 ** | 0.16 * | 1 | | | | | | | | |
| 10. | 0.16 | | 0.02 | −0.02 | 0.13 * | 0.17 ** | 0.06 | 0.11 | 0.02 | 0.01 | −0.03 | 1 | | | | | | | |
| 11. | 41.67 | 12.39 | −0.09 | 0.15 * | −0.22 ** | −0.12 | −0.14 * | −0.08 | 0.00 | −0.12 | 0.08 | −0.13 * | 1 | | | | | | |
| 12. | 0.51 | | 0.03 | 0.03 | 0.37 ** | 0.15 * | 0.05 | 0.07 | 0.09 | 0.15 * | −0.20 ** | 0.01 | −0.17 ** | 1 | | | | | |
| 13. | 5.22 | | 0.03 | −0.08 | 0.14 * | 0.02 | 0.14 * | −0.03 | 0.06 | 0.02 | −0.07 | 0.03 | −0.14 * | 0.08 | 1 | | | | |
| 14. | 4.19 | 1.84 | −0.03 | 0.06 | 0.26 ** | −0.12 | 0.13 * | 0.15 * | 0.15 * | −0.12 | 0.24 ** | 0.09 | 0.10 | 0.04 | 0.09 | 1 | | | |
| 15. | 0.75 | | 0.03 | 0.05 | −0.02 | 0.06 | 0.01 | 0.05 | 0.05 | 0.06 | −0.01 | 0.05 | 0.07 | 0.03 | 0.05 | 0.41 ** | 1 | | |
| 16. | 0.45 | | −0.13 * | 0.09 | 0.10 | 0.02 | 0.18 ** | 0.09 | 0.10 | 0.02 | 0.14 * | −0.00 | −0.05 | 0.10 | −0.07 | 0.33 ** | 0.22 ** | 1 | |
| 17. | 0.55 | | 0.09 | −0.12 | 0.14 * | 0.01 | 0.11 | 0.10 | −0.03 | 0.01 | −0.12 | −0.02 | −0.14 * | 0.24 ** | 0.10 | −0.10 | −0.05 | −0.02 | 1 |

Note A: * $p \leq 0.05$; ** $p \leq 0.01$. Note B: M—mean or proportion; SD—standard deviation; 1. = intention to quit; 2. = work engagement; 3. = teleworking; 4. = skill utilization; 5. = decision authority; 6. = workload; 7. = social support; 8. = recognition; 9. = number of hours worked; 10. = irregular work schedule; 11.= age; 12. = gender; 13. = educational level; 14. = household income; 15. = marital status; 16. = parental status; 17. = stress related to COVID-19.

**Table 2.** Main results.

| Hypotheses | Constructs and Direct Paths | Constructs and Indirect Paths | Unstandardized Beta | Inference |
|---|---|---|---|---|
| 1 | **Teleworking → Skill utilization** | | **1.242 \*\*** | **Partially supported** |
| | Teleworking → Decision authority | | 0.102 | |
| | Teleworking → Workload | | 0.245 | |
| | **Teleworking → Social support** | | **1.625 \*\*** | |
| | Teleworking → Recognition | | 0.775 | |
| | **Teleworking → Number of hours worked** | | **−2.604 \*\*** | |
| | Teleworking → Irregular work schedule | | 0.087 | |
| 2 | | **Teleworking → Work engagement (via Skill utilization)** | **0.904 \*** | **Partially supported** |
| | | Teleworking → Work engagement (via Decision authority) | 0.010 | |
| | | Teleworking → Work engagement (via Workload) | −0.050 | |
| | | Teleworking → Work engagement (via Social support) | 0.290 | |
| | | Teleworking → Work engagement (via Recognition) | 0.330 | |
| | | Teleworking → Work engagement (via Number of hours worked) | −0.286 | |
| | | Teleworking → Work engagement (via Irregular work schedule) | −0.051 | |
| 3 | | **Teleworking → Intention to quit (via Skill utilization and Work Engagement)** | **−0.283 \*** | **Partially supported** |
| | | Teleworking → Intention to quit (via Decision authority and Work engagement) | −0.003 | |
| | | Teleworking → Intention to quit (via Workload and Work engagement) | 0.016 | |
| | | Teleworking → Intention to quit (via Social support and Work engagement) | −0.091 | |
| | | Teleworking → Intention to quit (via Recognition and Work engagement) | −0.103 | |
| | | Teleworking → Intention to quit (via Number of hours worked and Work engagement) | 0.090 | |
| | | Teleworking → Intention to quit (via Irregular work schedule and Work engagement) | 0.016 | |
| | **ADJUSTMENTS** | | | |
| CFI | | | 0.997 | |
| | | | 0.956 | |
| | | | 597.920 (108) \*\* | |

Note A: $*$ $p \leq 0.05$; $**$ $p \leq 0.01$. Note B: The list of covariates: age, gender, educational level, household income, marital status, parental status, and stress related to COVID-19. See significant results in bold.

## 4. Discussion

The aims of this research were to investigate the direct effects of teleworking (i.e., teleworking from home) on work organization conditions and to verify the indirect impacts of TW on WE and IQ via work organization conditions. The sample included 254 French Canadian employees from 19 small and medium organizations (SMOs). Identification of the underlying mechanisms that link teleworking to work engagement and intention to quit is crucial to improving our understanding and ensuring the positive effects on employees, which would also be beneficial for the organizations that employ them.

Our first hypothesis (H1), which presumed that teleworking was directly associated with work organization conditions, was partially supported. Considering that the effects of teleworking appeared to be quite equivocal, we did not formulate specific hypotheses but rather general ones. Instead, we wanted to see the effects of teleworking on these variables in an exploratory way. We found that teleworking was directly associated with higher levels of skill utilization and social support and with a lower number of hours worked. Still, the results referring to the impact of teleworking on decision authority, workload, recognition, and irregular work schedules were insignificant. Regarding the effects on skill utilization and the number of hours worked, the obtained results are consistent with past empirical research. Indeed, it was reported that teleworking was associated with higher job decision latitude (i.e., autonomy and skill development) [29] and that it improved work schedules by allowing more comfortable workloads, reduced working hours, a compressed work week, and flexible shift systems [34]. The results pertaining to social support were opposite those from a previous study, in which employees teleworking more days a week reported a lower level of social support from their colleagues [32]. Moreover, we were surprised to find that our results did not reveal a significant effect of teleworking on several work organization conditions. Although we did not find past research that verified how recognition might be affected by teleworking, it was recently demonstrated that recognition was passed on or noticed more easily in the official workplace compared to when teleworking [33]. Nevertheless, our results clarify the effect of teleworking on work organization conditions while being aligned with the proposed extension of the JD-R model. Indeed, it appears that teleworking is an upstream resource that could impact other demands and resources at work, as it boosts skill utilization and social support while lowering the number of hours worked.

Our second hypothesis (H2), which postulated that teleworking was indirectly associated with work engagement via work organization conditions, was partially supported. Results indicated that teleworking was positively associated with work engagement via its effect on skill utilization. However, teleworking was not significantly associated with work engagement through its effects on decision authority, workload, social support, recognition, an irregular work schedule, or the number of hours worked. The direct effect of teleworking on skill utilization was still strong enough to indirectly influence work engagement, but its effects on social support and the number of hours worked were not strong enough to indirectly influence work engagement. This was unanticipated since it seems that the effect of teleworking on skill utilization is opposite of the direct effect previously identified linking teleworking and work engagement. In fact, it has previously been demonstrated that teleworking is associated with lower work engagement [9]. Note that this was also observed and reported in the results section of this study without being the object of a specific hypothesis. Thus, this seems to support the importance of the underlying mechanisms when discussing the impacts of teleworking on work engagement and helps to fine-tune the JD-R model [26]. Indeed, this study demonstrated that skill utilization, a powerful mediator, seems to partly explain the link between these variables. Therefore, an upstream resource is added to the traditional psychological process (the health impairment hypothesis) in the JD-R model [26]. Through this new process, teleworking, an upstream resource, influences work engagement through its effect on another organizational resource, namely, skill utilization. This indicates that if organizations ensure that teleworking provides employees with opportunities to use their skills, it will have beneficial rather than detrimental effects on their work engagement. An alternative explanation of this surprising result could be that employees who experience a high level of skill utilization hold different types of jobs that could be more resourceful, allowing more autonomy to work without constant supervision. Hence, employees experiencing a high level of skill utilization may, in general, have jobs better suited for teleworking compared to other employees.

Our third hypothesis (H3), which presumed that teleworking was indirectly associated with the intention to quit via its effect on work organization conditions and work engagement, was partially supported. We established that TW was associated with a

lower IQ via its effects on skill utilization and WE. However, TW was not significantly linked with intention to quit via its effects on decision authority, workload, social support, recognition, an irregular work schedule, weekly working hours, or work engagement. In other words, the effect that TW had on skill utilization was strong enough to indirectly influence WE and IQ. Again, the direct impacts of TW on social support as well as on the number of hours worked were not strong enough to indirectly influence WE and IQ. As for the second hypothesis, this result is surprising since it seems that the effect of teleworking on skill utilization reverses the indirect effect previously identified linking teleworking and intention to quit through work engagement [9]. Note that this was also observed and reported in the results section of this study without being the object of a specific hypothesis. The same reflection presented for the second hypothesis applies here, that is, the underlying mechanisms linking teleworking to intention to quit via work engagement make a major difference and enrich the JD-R model [26]. Indeed, skill utilization appears to be central to understanding the impacts of TW on those outcomes. As mentioned above, an upstream resource is added to another traditional psychological process (the motivational hypothesis) of the JD-R model [26]. Through this new process, teleworking, an upstream resource, influences work engagement and the intention to quit through its impact on skill utilization. Ultimately, this also confirms that teleworking helps to prevent a loss spiral in the context of COR theory [35] by influencing other work organization conditions and their effects on work engagement and the intention to quit.

Overall, we found that skill utilization was a powerful mediator of the association between teleworking and work engagement, which also affects the intention to quit. Teleworking was also directly associated with higher skill utilization and social support, as well as a lower number of hours worked. Accordingly, teleworking and skill utilization are important work organization conditions (also termed organizational resources) for employers that aspire to increase employees' work engagement and decrease their intention to quit in the actual chronic labor shortage context.

### 4.1. Practical Implications

Although the results show that teleworking is only significantly associated with skill utilization, social support, and the number of hours worked, organizations should not underestimate its potential positive impact on other outcomes or work organization conditions. It is particularly important to consider that this study was carried out with a sample of employees from 19 SMOs. Their experience, which might be characterized by constant interactions, proximal management style, spatial closeness [24], as well as omnipresent communications [25], does not necessarily apply to that of the general workforce. In fact, the results reveal that teleworking should be considered a way to increase employees' work engagement and intention to quit due to its effect on skill utilization in these 19 SMOs (H1, H2, and H3). It is possible that the size of these organizations results in more closeness between employees and managers (and more social support), even at a distance, and this could foster the positive influence of teleworking on skill utilization, which successively impacts work engagement and the intention to quit. Additionally, it must be noted that autonomy (e.g., skill utilization) could be a double-edged sword [64], as it can increase job complexity and responsibility [65]. As such, organizations should invest in the implementation of teleworking, seeking to ensure that it allows for adequate but balanced autonomy on the part of employees (e.g., skill utilization). This would likely involve avoiding micromanagement (i.e., closely observing and controlling the work details of employees) when implementing teleworking in organizations, as such management can harm the autonomy of employees [66,67] and thus have consequences for their work engagement and intention to quit. Employee autonomy also needs to be balanced with a good organizational support system, for example, involving colleagues and supervisors. In the same vein, organizations and their managers could prioritize performance management based on the achievement of objectives rather than attempting to control the number of hours worked or implementing strict working schedules. In other words, when teleworking, employees' objectives must be

evidently articulate, and organizations and managers need to have faith in them [68]. In line with this notion, a recent study showed that results-based management and trust-building actions improve workplace performance in the context of teleworking [69]. Of course, it is also important to observe good general organizational practices surrounding teleworking, such as reaching out to employees every day to preserve social interactions [70] and ensuring sustained dialogue regarding requirements and task advancement [71]. Finally, organizations must focus on skill utilization, which is an important work organization condition, as well as emphasizing employees' strengths [72] without giving them too much responsibility to avoid the possible downside of autonomy, as previously discussed.

### 4.2. Limitations and Future Directions

This research has limitations. First, this research is cross-sectional, making it impossible to establish causality between variables. Second, another limitation of this study concerns the small sample size (i.e., 254 employees). Third, there is a possibility of common variance bias because all the variables were collected from the same source. However, this bias should be considered low due to the diversity of our sample organizations ($n = 19$), which were from both secondary and tertiary economic sectors, only some of which had unions. Fourth, we used single-source data in the form of a self-reported set of questions filed by volunteer employees, which can lead to biases in the perception and understanding of the questions in addition to social desirability issues. However, the respondents were advised that their responses were completely anonymous. Conducting interviews with supervisors as well could have limited response bias. Fifth, we believe that it is essential to highlight the potentiality of selection bias caused by the workplaces that participated in the study. These workplaces are certainly more aware of the importance of their employees and more willing to do their best to enrich their organizational practices for their benefit. Sixth, considering the previous issues, the results of this study may not be generalizable to all organizations. Seventh, even though gender and age were controlled in this study, teleworking could have differential impacts for men and women and for different age groups. Additionally, race and ethnicity were not measures in this research. Accordingly, the associations with gender, age, and race and ethnicity need to be verified in the future. In addition, longitudinal studies are required in the future, as there is currently no clear information regarding the chronic effects of teleworking over time. The different teleworking configurations (e.g., number of days a week) should also be considered in future research to determine whether there is an optimal number of days of work per week to ensure the proper functioning of employees. Additionally, future studies should verify the effect of imposed teleworking. Even though we assume that teleworking was imposed on most of our respondents in the context of the COVID-19 pandemic, we cannot be certain that this was the case for all of them. Furthermore, it is very likely that not all jobs are equally suitable for teleworking, depending on the tasks they require. This might influence the associations between teleworking, work engagement, and the intention to quit. In the same vein, home conditions, such as space available for teleworking and having kids at home, might influence work engagement when teleworking. Accordingly, future studies should consider the possible confounding effect those variables may have on these associations. In addition, other factors that could influence the effects of teleworking and its optimal configuration on various outcomes should be considered in future studies. The optimal configuration of teleworking may vary depending on the individual characteristics of the employees, such as their personality traits, their level of emotional intelligence, and their lifestyle habits. From an inclusive perspective, it would be useful to verify whether teleworking can favor employees with mobility challenges. Indeed, the benefits of teleworking could be viewed more broadly and not only in terms of work–family balance (for employees who are also parents). Teleworking can also support the inclusion of alternative (but not necessarily mutually exclusive) categories of employees by offering them the flexibility they need; for example, individuals with chronic diseases that limit their physical ability to travel to work. Some of these physical disabilities do not necessarily require a leave of absence, whether

temporary or not, but rather accommodation. This might, for instance, include employees who suffer from severe anemia but who nevertheless have the intellectual and mental abilities to work well. Without the possibility of teleworking, a leave of absence would be necessary for an extended period of time. It could also include employees suffering from pain, chronic fatigue, or obesity. Thus, teleworking can support diversity and inclusion, and this is an issue that could be considered in future studies. Finally, it is essential to replicate this study with alternative samples in respect of workplaces (not only small and medium-sized), workers, and other countries.

## 5. Conclusions

This research's central aims were to investigate the direct impacts of TW on work organization conditions as well as its indirect effects on WE and IQ through work organization conditions. A few key findings emerged. Teleworking appears to be an upstream resource that has the potential to impact other resources at work (e.g., work organization conditions) in SMOs. Further, teleworking also appears to influence work engagement and the intention to quit differently and in a more positive way in terms of its effect on skill utilization. This highlights the importance of the underlying mechanisms linking teleworking with work engagement and the intention to quit. Therefore, some trajectories of teleworking have been highlighted in this cross-sectional research based on the theoretical model developed. Accordingly, teleworking must be designed to give employees the opportunity to use their skills, which in turn will have additional beneficial effects. The results of this study also support refining the JD-R model by adding to the usual dual processes that lead to the development of health impairment and motivation. While acknowledging the intrinsic limitations of this research, we trust that it will play a part in the ongoing discussions regarding teleworking. The results obtained in this research widen the available empirical knowledge by demonstrating the extended trajectories via which TW is connected to WE and IQ and by providing distinct practical implications as well as future research directions. Obviously, relying on the implementation of optimal and sustainable teleworking and skill utilization in the context of the present chronic labor shortage is only part of the solution for organizations. It will also be necessary to develop alternative solutions, such as immigration, tax incentives for retirees returning to work, and better recognition of foreign diplomas.

**Author Contributions:** Conceptualization, A.P.-L.; Methodology, A.P.-L.; Software, A.P.-L.; Formal analysis, A.P.-L. and A.M.; Investigation, A.P.-L.; Data curation, A.P.-L.; Writing—original draft, A.P.-L.; Writing—review & editing, A.M.; Funding acquisition, A.P.-L. All authors have read and agreed to the published version of the manuscript.

**Funding:** This research was funded by Social Sciences and Humanities Research Council [grant number 430-2020-0674]; Fonds de Recherche du Québec - Société et culture [grant number 267581]; as well as the UQTR Junior Research Chair on HRM practices, Well-being, and Performance at Work.

**Institutional Review Board Statement:** The study was conducted in accordance with the Declaration of Helsinki, and approved by the Institutional Review Board of the Université du Québec à Trois-Rivières (protocol code: CER-20-270-08-02.37; date of approval: 6 January 2020).

**Informed Consent Statement:** Informed consent was obtained from all subjects involved in the study.

**Data Availability Statement:** The data are not publicly available in order to respect the privacy of research participants.

**Conflicts of Interest:** The authors declare no conflict of interest.

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
