# Peer review of "Trajectories of Teleworking via Work Organization Conditions: Unraveling the Effect on Work Engagement and Intention to Quit with Path Analyses"

_sustainability, doi:10.3390/su15118476_

Round 1
Reviewer 1 Report
I found the article clear and up to date. The methodology is acceptable and the results are clearly presented. The article contributes to the science, although it has limitations, most notably in the number and structure of respondents.
Author Response
REVIEWER 1:
I found the article clear and up to date. The methodology is acceptable and the results are clearly presented. The article contributes to the science, although it has limitations, most notably in the number and structure of respondents.
Thank you, reviewer, 1 for your positive feedback.
Reviewer 2 Report
General comments: This is an interesting and well-written paper on the topic of teleworking (TW). TW has increased dramatically in many countries after Covid-19, and there is need for more knowledge on how TW impacts health and wellbeing from the perspective of the employees and job turnover, absenteeism etc. from the perspective of the workplace. The study in the paper sheds more light on these important aspects. The study focus on well-known psychosocial working conditions (workload, support, etc.), and uses the job demands/resources framework to build a model of the connections between TW and work engagement (WE) and intention-to-quit (ITQ). However, the relation between TW and WE and ITQ is probably very complex and depends on many other factors in addition to the work conditions addressed in the study. For example, very likely not all jobs are equally suitable for teleworking, and therefore your job and specific work tasks may influence the association between TW and WE. I think that possible “confounding” by job and work tasks deserves to be addressed in the paper. Also, home conditions (space available for TW, parenting of kids) might influence WE when TW. See also specific comments to the discussion.
Specific comments:
Title: Is ‘trajectories’ the right word here? I am not a native English speaker, therefore I am not sure. But ´trajectories’ makes me think of something time-dependent. But since the study is cross-sectional, this resonates not so well with ‘trajectories’. Consider using another word.
Abstract: Although the direct associations of TW with WE and ITQ was not a hypothesis, I think it should mention that TW is negatively associated with WE. It is an important aspect of TW if it – everything else being equal – contributes to lower WE. Perhaps that the meaning of the sentence in line 28, but is not very clear what is meant by “…teleworking can be harmful…”. If this sentence refers to the results, you should rephrase it, so it is more specific.
p. 3, l. 114-116: I am not convinced that employees in SMO’s are more ‘habituated to interact more frequently’ and ‘share spatial space’ compared to other organizations. It depends on the type of job, I guess. Have you any reference to support the postulate? Also, see my comments to "Procedures and participants" below.
p.3, l. 148-150: Unclear. I have read the sentence over and over, and I still don’t understand what it says.
p.4 Procedures and participants: Can you be more specific on the type of work carried out by the participants?
p.4 Procedures and participants: If you have the information: Was TW mandatory or optional for the TW employees?
p.4 Procedures and participants: You mention that the organizations were from the secondary and tertiary economic activity sector. Can you be more specific on the type of business that are represented in the sample?
p.4 Procedures and participants: I recommend to adjust the numbers to just one decimal (for example, 36.84% à 36.8%). The precision expressed by two decimals is excessive in this case and makes reading harder.
p.4, l. 209: Some words must be missing from the sentence. Or perhaps the first two words, ‘This research’ are superfluous.
p.7, results: In the text you report that TW was significantly associated with a lower level of WE. However, in Table 1 the correlation is between TW and WE low (-0.08) and non-significant. I think you should comment on this discrepancy.
p. 9, l. 328. I assume that ‘constant’à’consistent’
p. 9-11 Discussion. An overall finding is the important role of skill utilization of mediating the effect of TW on WE and ITQ. I don’t disagree with the points made in discussion, but in light of the cross-sectional nature of the data, I would like add a critical note. For example, an alternative explanation of the results could be that, as a hypothesis, that employees that experience a high skill utilization hold different types of jobs compared other employees, and they could be more resourceful, independent and able to work without constant supervision. Hence, employees experiencing a high skill utilization may in general have jobs better suited for TW compared to other employees (still, as a hypothesis). Thus, the results may reflect that TW is better suited for some type of jobs.
p. 11. Limitations and recommendations for future research: The authors should mention the limitation of the cross-sectional study design. Only associations, not causality or causal directions, can be inferred from cross-sectional data.
p. 11. Limitations and recommendations for future research: Another limitation is the number of participants. N=254 is not many and puts a restriction on more advanced analyses, for example, moderation analysis (analysis of interactions) and adjustment for the effects of possible confounders.
Author Response
REVIEWER 2:
General comments: This is an interesting and well-written paper on the topic of teleworking (TW). TW has increased dramatically in many countries after Covid-19, and there is need for more knowledge on how TW impacts health and wellbeing from the perspective of the employees and job turnover, absenteeism etc. from the perspective of the workplace. The study in the paper sheds more light on these important aspects. The study focus on well-known psychosocial working conditions (workload, support, etc.), and uses the job demands/resources framework to build a model of the connections between TW and work engagement (WE) and intention-to-quit (ITQ).
RESPONSE:
Thank you for your positive feedback.
However, the relation between TW and WE and ITQ is probably very complex and depends on many other factors in addition to the work conditions addressed in the study. For example, very likely not all jobs are equally suitable for teleworking, and therefore your job and specific work tasks may influence the association between TW and WE. I think that possible “confounding” by job and work tasks deserves to be addressed in the paper. Also, home conditions (space available for TW, parenting of kids) might influence WE when TW. See also specific comments to the discussion.
RESPONSE:
Thank you for your valuable suggestions. Accordingly, we added it in the limitations section. See track changes on page 11 of the revised manuscript.
Specific comments:
Title: Is ‘trajectories’ the right word here? I am not a native English speaker, therefore I am not sure. But ´trajectories’ makes me think of something time-dependent. But since the study is cross-sectional, this resonates not so well with ‘trajectories’. Consider using another word.
RESPONSE:
We fully understand your point of view and believe that it is true that the word trajectories may seem to intend the notion of time. However, our objective was to signify the notion of progression not in terms of time, but in terms of a theorised sequence. In support of this sequence/path, is our theoretical model which posits that teleworking might influence other work organization conditions. In fact, teleworking could be an upstream demand or a resource that could impact other demands and resources at work. Recent studies seem to indicate that teleworking is conducting to an alteration of the work organization conditions. However, and to avoid confusion, we took care to specify this in the conclusions of our manuscript (See track changes on page 12 of the revised manuscript). Furthermore, we have clearly indicated the fact that this research was cross-sectional and have also added this information in the limitations (See track changes on page 11). We hope this will answer your very relevant question. We find this word better suit to represents our theoretical model and find it difficult to find a better word that would reflects the path analyses that were carried out. The direct and indirect impacts represent the trajectories. Thank you very much.
Abstract: Although the direct associations of TW with WE and ITQ was not a hypothesis, I think it should mention that TW is negatively associated with WE. It is an important aspect of TW if it – everything else being equal – contributes to lower WE. Perhaps that the meaning of the sentence in line 28, but is not very clear what is meant by “…teleworking can be harmful…”. If this sentence refers to the results, you should rephrase it, so it is more specific.
RESPONSE:
Thank you for your important suggestion. Accordingly, we were more specific in that sentence. See track changes in the Abstract.
- 3, l. 114-116: I am not convinced that employees in SMO’s are more ‘habituated to interact more frequently’ and ‘share spatial space’ compared to other organizations. It depends on the type of job, I guess. Have you any reference to support the postulate? Also, see my comments to "Procedures and participants" below.
RESPONSE:
Thank you for your valuable question. In fact, we had a reference supporting this statement (See reference below). That said and in accordance with your question, we added that this might also depend on the type of job. See track changes on page 3 of the revised manuscript.
Reference:
Torrès, O., Petitesse des entreprises et grossissement des effets de proximité. Revue française de gestion 2015, 41, (253), 333-352.
p.3, l. 148-150: Unclear. I have read the sentence over and over, and I still don’t understand what it says.
RESPONSE:
Thank you for bringing this to our attention. Accordingly, we corrected the sentence. See track changes on pages 3-4 of the revised manuscript.
p.4 Procedures and participants: Can you be more specific on the type of work carried out by the participants?
RESPONSE:
Thank you for your very relevant question. Unfortunately, we did not measure the type of job in our questionnaire. However, we have indicated it in the limitations and suggestions for future research.
p.4 Procedures and participants: If you have the information: Was TW mandatory or optional for the TW employees?
RESPONSE:
Thank you for your very relevant question. Unfortunately, we do not have this information. We did not ask that question to the participants. That said, we believe that it might have been imposed for most of them since data were collected during the pandemic. But we cannot confirm it per se. Therefore, we added it in the limitations (See track changes on page 11).
p.4 Procedures and participants: You mention that the organizations were from the secondary and tertiary economic activity sector. Can you be more specific on the type of business that are represented in the sample?
RESPONSE:
Thank you for your valuable question. Accordingly, we added two examples in the revised version of the manuscript (See track changes on page 5). But for confidentiality reasons, this corresponds to the most precise information that we can give about the sectors of activity of the participating organizations.
p.4 Procedures and participants: I recommend to adjust the numbers to just one decimal (for example, 36.84% à 36.8%). The precision expressed by two decimals is excessive in this case and makes reading harder.
RESPONSE:
Thank you for your recommendation. We made the changes accordingly. See track changes on page 5 of the revised manuscript.
p.4, l. 209: Some words must be missing from the sentence. Or perhaps the first two words, ‘This research’ are superfluous.
RESPONSE:
Thank you very much. Accordingly, we rephrase that sentence. See track changes on page 5 of the revised manuscript.
p.7, results: In the text you report that TW was significantly associated with a lower level of WE. However, in Table 1 the correlation is between TW and WE low (-0.08) and non-significant. I think you should comment on this discrepancy.
RESPONSE:
Thank you for your valuable question. Table 1 showed bivariate correlations. This type of analysis does not consider the effect of other variables present in the model, as our main model does. Regression analyses thus revealed a suppressive effect that make the association between WE and TW statistically significant when adjusted for other variables in the model.
- 9, l. 328. I assume that ‘constant’à’consistent’
RESPONSE:
Thank you. Indeed, that what was we met. We made the correction accordingly. See track changes on page 8 of the revised manuscript.
- 9-11 Discussion. An overall finding is the important role of skill utilization of mediating the effect of TW on WE and ITQ. I don’t disagree with the points made in discussion, but in light of the cross-sectional nature of the data, I would like add a critical note. For example, an alternative explanation of the results could be that, as a hypothesis, that employees that experience a high skill utilization hold different types of jobs compared other employees, and they could be more resourceful, independent and able to work without constant supervision. Hence, employees experiencing a high skill utilization may in general have jobs better suited for TW compared to other employees (still, as a hypothesis). Thus, the results may reflect that TW is better suited for some type of jobs.
RESPONSE:
Many thanks for this very interesting alternative explanation. Accordingly, we added it on pages 9-10 of the revised manuscript. See track changes.
- 11. Limitations and recommendations for future research: The authors should mention the limitation of the cross-sectional study design. Only associations, not causality or causal directions, can be inferred from cross-sectional data.
RESPONSE:
Thank you for your pertinent remark. Accordingly, we added is as a limitation. See track changes on page 11 of the revised manuscript.
- 11. Limitations and recommendations for future research: Another limitation is the number of participants. N=254 is not many and puts a restriction on more advanced analyses, for example, moderation analysis (analysis of interactions) and adjustment for the effects of possible confounders.
RESPONSE:
Thank you for your pertinent remark. Accordingly, we added is as a limitation. See track changes on page 11 of the revised manuscript.
We would like to thank reviewer 2 for their comments that helped improve our manuscript.
Reviewer 3 Report
Research design in this article lacks clarity. The researchers need to more clearly define what they did in order for the reader to understand. Revisions to improve clarity of study procedures appear warranted. Moreover, present research questions and their associated hypotheses together.
Please clearly connect analyses to the research questions and/or hypotheses they seek to answer.
Additionally, provide details in relation with participants. Please include additional demographic information related to socio-economic status, race-ethnicity, etc. if available. Additional explanation and description should be provided for each of the measures included in the study. Please include psychometric evidence supporting use of these measures if available. If unavailable, please provide a rationale for their use.
Please reorganize content into “Results” and “Discussion” sections. In each section, clearly connect results and discussion content to the research question or hypothesis they address.
Implications needed to be clearly presented and discussed in relation with the opted study derived results.
Please review the article for grammar/syntax errors. Please review the article for APA errors. APA formatting rules (e.g., citations, use of parentheses, manuscript sections/headings/subheadings). Please conform for more conventional APA recommended headings and subheadings (i.e., introduction, Method [Participants, Measures, Procedures, Analysis Plan], Results, Discussion, Limitations and Future Directions, Conclusions).
Author Response
REVIEWER 3:
Research design in this article lacks clarity. The researchers need to more clearly define what they did in order for the reader to understand. Revisions to improve clarity of study procedures appear warranted. Moreover, present research questions and their associated hypotheses together. Please clearly connect analyses to the research questions and/or hypotheses they seek to answer.
RESPONSE:
Thank you for your important recommendation. Accordingly, we clearly connected analyses to the hypotheses. See track changes on page 7 of the revised manuscript.
Additionally, provide details in relation with participants. Please include additional demographic information related to socio-economic status, race-ethnicity, etc. if available. Additional explanation and description should be provided for each of the measures included in the study. Please include psychometric evidence supporting use of these measures if available. If unavailable, please provide a rationale for their use.
RESPONSE:
Thank you for your suggestions. Accordingly, we added information related to the household income of our participants on page 5 of the revised manuscript (See track changes). Unfortunately, our questionnaire did not measure the race-ethnicity of participants. Accordingly, we added it as a limitation and suggestion for future research on page 11 of the revised manuscript (See track changes). Also, note that our measures were all previously validated and that we gave the references in the manuscript (we highlighted in red to make it more visible), except for the measurement of teleworking and stress related to the COVID-19 pandemic. The fact that they are validated measures widely use in the literature serve as a rationale for their use in this research. Regarding teleworking and stress related to COVID-19 pandemic, our rationale was to measure if those employees were teleworking or not and if they were stressed because of the COVID-19 pandemic or not. Therefore, we ask those single questions and treated the variables as dichotomous. That said, what could have been done was to measure whether telework was imposed or not. Even though we assume that this was imposed in the context of a pandemic, we cannot be certain that this was the case for all these employees. Consequently, we have added this in the limitations and suggestions for future research (See track changes on page 11 of the revised manuscript).
Please reorganize content into “Results” and “Discussion” sections. In each section, clearly connect results and discussion content to the research question or hypothesis they address.
RESPONSE:
Thank you for your pertinent suggestion. Accordingly, we connected our results section with the hypothesis they addressed. See track changes on page 7 of the revised manuscript. In our discussion section, hypotheses were already connected in the first version of the manuscript. We highlighted in red to make it more visible.
Implications needed to be clearly presented and discussed in relation with the opted study derived results.
RESPONSE:
Thank you for this valuable recommendation. Accordingly, we connected the implications with the results and hypothesis they derived from. See track changes on page 11 of the revised manuscript. In fact, all our practical suggestions are related to the importance of skill utilization/autonomy (which is a related concept). Additionally, we added some good general organizational practices surrounding teleworking to make sure that it remains positive for employees. All our suggestions derive from the literature and are important to consider in relation with our results that mainly point to the importance of skill utilization.
Please review the article for grammar/syntax errors. Please review the article for APA errors. APA formatting rules (e.g., citations, use of parentheses, manuscript sections/headings/subheadings). Please conform for more conventional APA recommended headings and subheadings (i.e., introduction, Method [Participants, Measures, Procedures, Analysis Plan], Results, Discussion, Limitations and Future Directions, Conclusions).
RESPONSE:
Prior to the first submission, we carried out a paid linguistic revision by a professional. We noticed that there were still some errors which we have corrected in this revised version. If necessary, we are ready to redo the paid linguistic revision by a professional. Thank you very much for your advice.
We would like to thank reviewer 3 for their comments that were instrumental in improving the quality of our manuscript.